# Accuracy of Computer-Assisted Surgery in Maxillary Reconstruction: A Systematic Review

**DOI:** 10.3390/jcm10061226

**Published:** 2021-03-16

**Authors:** Gustaaf J. C. van Baar, Kitty Schipper, Tymour Forouzanfar, Lars Leeuwrik, Henri A. H. Winters, Angela Ridwan-Pramana, Frank K. J. Leusink

**Affiliations:** 1Department of Oral and Maxillofacial Surgery/Pathology, Amsterdam UMC and Academic Center for Dentistry Amsterdam (ACTA), Vrije Universiteit Amsterdam, De Boelelaan 1117, 1081 HV Amsterdam, The Netherlands; k.schipper1@amsterdamumc.nl (K.S.); t.forouzanfar@amsterdamumc.nl (T.F.); l.leeuwrik@amsterdamumc.nl (L.L.); a.ridwan@amsterdamumc.nl (A.R.-P.); f.leusink@amsterdamumc.nl (F.K.J.L.); 2Department of Plastic, Reconstructive and Hand Surgery, Amsterdam UMC, Vrije Universiteit Amsterdam, De Boelelaan 1117, 1081 HV Amsterdam, The Netherlands; h.winters@amsterdamumc.nl; 3Centre for Special Care in Dentistry, Department of Maxillofacial Prosthodontics, Vrije Universiteit Amsterdam, De Boelelaan 1117, 1081 HV Amsterdam, The Netherlands

**Keywords:** maxillofacial reconstruction, free tissue flaps, surgery, computer-assisted, computer-aided design, computer-aided manufacturing, accuracy

## Abstract

Computer-assisted surgery (CAS) in maxillary reconstruction has proven its value regarding more predictable postoperative results. However, the accuracy evaluation methods differ between studies, and no meta-analysis has been performed yet. A systematic review was performed in the PubMed, Embase, and Cochrane Library databases, using a Patient, Intervention, Comparison and Outcome (PICO) method: (P) patients in need of maxillary reconstruction using free osteocutaneous tissue transfer, (I) reconstructed according to a virtual plan in CAS software, (C) compared to the actual postoperative result, and (O) postoperatively measured by a quantitative accuracy assessment) search strategy, and was reported according to the PRISMA statement. We reviewed all of the studies that quantitatively assessed the accuracy of maxillary reconstructions using CAS. Twelve studies matched the inclusion criteria, reporting 67 maxillary reconstructions. All of the included studies compared postoperative 3D models to preoperative 3D models (revised to the virtual plan). The postoperative accuracy measurements mainly focused on the position of the fibular bony segments. Only approximate comparisons of postoperative accuracy between studies were feasible because of small differences in the postoperative measurement methods; the accuracy of the bony segment positioning ranged between 0.44 mm and 7.8 mm, and between 2.90° and 6.96°. A postoperative evaluation guideline to create uniformity in evaluation methods needs to be considered so as to allow for valid comparisons of postoperative results and to facilitate meta-analyses in the future. With the proper validation of the postoperative results, future research might explore more definitive evidence regarding the management and superiority of CAS in maxillary and midface reconstruction.

## 1. Introduction

Ablative surgery of the maxilla can lead to a high level of functional impairment, and often causes a psychological and aesthetical trauma for the patient. Traditionally, large maxillary defects were treated by a prosthetic obturator [1]. Currently, free osteocutaneous tissue transfer is the primary method of choice for large defects [2], permanently separating the oral and nasal cavities, with both physical and psychological benefits for the patient [3]. The donor sites described in the literature are mainly the fibula free flap (FFF), followed by the scapular osteocutaneous free flap (SOFF) and the deep circumflex iliac artery flap (DCIA) [2,4,5].

The goals of maxillary reconstruction are defect obliteration, support of midfacial elements, restoration of oral and nasal function, and restoration of facial aesthetics [6].

To achieve these goals, the bony maxillary reconstruction needs to be accurate for two reasons. Firstly, a high accuracy is needed in cases involving orbital floor reconstruction for the correct positioning of the eyeball in order to prevent enophthalmos and or diplopia [7]. Secondly, high accuracy is essential when dental implants are placed during the reconstruction, in a guided fashion and in line with the antagonist dentition, in order to achieve optimal occlusion [8]. This shortens the total oral rehabilitation time (TORT), reducing the time and costs, and potentially improving quality of life [9].

With the introduction of computer-assisted surgery (CAS) software, reconstructive surgeons and thier medical engineers have the opportunity to virtually plan osteotomies of the resection and donor sites, mirror the remnant maxilla for contouring of the neomaxilla, create 3D medical devices for guided surgery, manufacture patient-specific reconstruction plates, optimize the positioning of dental implants, and restore correct occlusion [10]. Both preoperative virtual planning and the perioperative use of 3D devices result in more predictable positioning of the bony segments of the flap. In addition, CAS offers the possibility of primary guided dental implant placements, which reduces the number of surgical procedures and decreases the TORT [8,11] and costs [9]. Other benefits of CAS in maxillary reconstruction are a shorter ischemia and operative time compared with the conventional technique [11].

The accuracy evaluation of the postoperative bony maxillary reconstruction (including potential CAS guided dental implants) vs. the preoperative virtual plan provides the operator with information about the execution of the operation and gives an opportunity to compare the results with other techniques performed (CAS) in other studies. However, uniformity in accuracy evaluation methods is lacking, and no evaluation guideline like the one previously published for mandibular reconstruction using CAS exists for maxillary reconstructions as of yet [12]. Similar to our previously published systematic review on accuracy in mandibular reconstruction [13,14], we reviewed all studies that quantitatively assessed the accuracy of maxillary reconstruction performed with CAS as mentioned above. Based on the accuracy results in the mandibular reconstruction, we hypothesize that the postoperative deviation of the bony segments of the flap remains within 5 mm compared with the preoperative virtual plan.

## 2. Materials and Methods

This systematic review was conducted in accordance with the Preferred Reporting Items for Systematic Reviews and Meta-Analysis (PRISMA) statement [15]. The aim was to include all studies with a postoperative quantitative accuracy assessment of maxillary reconstructions performed with CAS, including studies comparing the accuracy results of CAS vs. the conventional technique. The PICO search strategy that was used was as follows: (P) patients in need of maxillary reconstruction using free osteocutaneous tissue transfer, (I) reconstructed according to a virtual plan in CAS software, (C) compared to the actual postoperative result, and (O) postoperatively measured by a quantitative accuracy assessment.

### 2.1. Search Strategy

A comprehensive search was performed in the PubMed, Embase, and Cochrane Library Scopus bibliographic databases, from inception until 8 January 2021, in collaboration with a medical librarian (L.S.). Search terms included controlled terms (MesH in PubMed and Emtree in Embase) and free text terms. In the Cochrane Library, we only used free text. The following terms were used (including synonyms and closely related words) as index terms or free-text words: “maxillary reconstruction”, “computer-assisted surgery”, and “accuracy”. A search filter was used to limit the results to humans and adults. The search was performed without date or language restrictions. Additionally, a manual search was performed in order to find additional relevant studies in the reference lists of the included studies. Duplicate articles were excluded. The full search strategies for all databases can be found in Appendix A.

### 2.2. Selection of Articles

Three independent reviewers (G.J.C.v.B., K.S., and L.L.) performed the selection of the relevant titles and abstracts. Randomized controlled trials, cohort studies (prospective observational), case-control studies, and case reports and series (either prospective or retrospective) were all considered applicable for this systematic review. The applied inclusion criteria were as follows: (1) maxillary reconstruction, (2) using CAS, and (3) including a postoperative accuracy assessment. The applied exclusion criteria were as follows: (1) non-osseous flaps or obturators; (2) presentation of mandible or skull base reconstruction data, with no ability to filter data pertaining to the maxilla; (3) no focus on reconstructive data; (4) no original research articles (e.g., abstract publications, editorials, letters, oral papers, or posters); and (5) extended maxillectomies.

### 2.3. Data Extraction

Included studies were screened for the following variables: author and year of publication, number of cases, primary vs. secondary reconstructions, defect size, flap type, number of segments, plate type, involvement of orbital floor reconstruction, pre- and postoperative imaging methods, CAS software type, use of a software mirroring tool, use of 3D printed devices, use of CAS guided dental implants, and the use of surgical navigation during the operation. The method of accuracy evaluation and the quantitative results were analysed. Data were extracted from the included studies and verified by three reviewers (G.J.C.v.B., K.S., and L.L.).

### 2.4. Risk of Bias

For assessing the risk of bias in the included case-series studies, the Institute of Health Economics (IHE) quality appraisal 20-criterion checklist (optimal score of 20) was applied [16]. For assessing the risk of bias in nonrandomized case-control studies, the methodological 12-criterion index for nonrandomized studies (MINORS; optimal score of 24) was applied [17]. High scores in both assessments corresponded to a higher quality and a lower risk of bias.

## 3. Results

### 3.1. Study Selection

Our systematic search generated a total of 2677 references—920 in PubMed, 1714 in Embase, and 43 in Cochrane Library. A total of 2014 unique titles remained after removing the duplicates. After screening the references for eligibility, 53 abstracts were selected and subsequently their full text was analyzed thoroughly. According to the inclusion and exclusion criteria, 12 studies were applicable to the purpose of this systematic review, describing a total of 67 maxillary reconstructions using CAS [8,11,18,19,20,21,22,23,24,25,26,27]. Figure 1 shows the PRISMA flowchart of the literature search and study selection process.

### 3.2. Study Characteristics

The 12 included studies described eight case series and four non-randomized case control series studies. The IHE appraisal scores (case series) and the MINORS scores (case-controlled series) of the included studies are shown in Appendix A. Case series studies are classified as low-quality evidence because of the absence of randomization, blinding, and a control group. The average IHE appraisal score of all case series together was 11.75 out of 20, which indicated a high risk of bias. The MINORS score of the non-randomized case control series was an average of 17 out of 24, which indicated a moderate risk of bias.

Table 1 shows the study characteristics, including 54 primary reconstructions and 13 secondary reconstructions. The Brown maxillary defect classification [2] was applied in 52 cases, and no defect classification was applied in 15 cases. Fifty-eight cases were reconstructed with an FFF and nine cases with a SOFF, and the number of bony segments varied between one and five. To fixate the bony segments, pre-bent mini-plates were used in 37 cases, mini-plates in 11 cases, a patient specific reconstruction plate (PSRP) in seven cases, and a patient specific titanium mesh in four cases. In the case of orbital floor reconstruction, a pre-bent titanium mesh was used in four cases and a 3D printed patient specific titanium mesh was used in two cases.

### 3.3. CAS Process

Table 2 shows the planning and modelling methods of the included studies. Preoperative craniofacial imaging was carried out using Computed Tomography (CT) in 48 cases, cone beam CT (CBCT) in 5 cases, and an unknown method in 14 cases. Preoperative donor site imaging was carried out using CT in 42 cases, CTA in 9 cases, and unknown in 16 cases. Postoperative craniofacial imaging was carried out using CT in 46 cases, CBCT in 6 cases, and unknown in 15 cases. The computer software that was used for the virtual planning of the maxillary reconstruction was Proplan/Surgicase CMF in forty-one cases, Mimics in nine cases, iPlan CMF in eight cases, Simplant Pro in five cases, 3-Matic in five cases, Rhino in four cases, InVesalius in four cases, Surfacer in two cases, and a free software format in one case. Swendseid et al. used either ProPlan CMF, Stryker CMF, or IPS Case Designer for their 9 VSP cases, but they did not mention the exact distribution of their software usage [26]. A mirroring tool for contralateral (unaffected) maxilla projection on the maxillary defect was used in 13 cases.

The printed 3D devices applied during the surgery phase were a cutting guide for the fibula in 48 cases, a cutting guide for the scapula in 9 cases, cutting guides for the maxilla in 56 cases, a shape template for the neomaxilla in 32 cases, a 3D model of the neomaxilla in 40 cases, a 3D model of the mirrored maxilla in 8 cases, a 3D printed titanium bridge abutment in five cases, a drill guide for dental implants in five cases, a 3D model of the native maxilla in four cases, a 3D model of the osteotomized maxilla in one cases, and a 3D model of the osteotomized fibula in one case.

Dental implants were included in the preoperative virtual planning of the maxillary reconstruction in six cases. Surgical navigation was used in 12 cases, of which eight cases used Brainlab software and five cases used Stryker software.

### 3.4. Evaluation Methods

Table 3 shows the measurement software used during the postoperative evaluation. Geomagic Studio was used in thirteen cases, ProPlan CMF in nine cases, Simplant combined with GOM in four cases, CloudCompare in four cases, Mimics in three cases, and unknown in thirty-four cases. In all cases, a postoperative STL model was compared to a preoperative STL model (revised to the virtual plan), and the postoperative accuracy primary outcome measurements mainly focused on the position of the bony segments (deviation (mm), shift (mm), dimension differences (mm), center point deviation (mm), or rotation (degrees)). The primary outcome measurements of the bony segments in the included studies ranged between 0.44 mm and 7.8 mm and between 2.90° and 6.96°.

In addition to the above-mentioned quantitative accuracy measurements, two studies determined the anatomical position of the bony segments compared with the pre-op anatomical situation or a mirrored reconstruction [26,27]. Both studies reported 100% of segments in the anatomical position in accordance with the planning.

In five cases, a postoperative accuracy measurement was performed for virtually planned dental implants. A mean center point deviation of 4.95 mm and a mean angulation deviation of 6.26° between the postoperative dental implant position and the preoperative virtual planned dental implant position was measured [8].

Three studies [21,22,24] reported about reconstruction of the orbital floor. None of these reported any quantitative measurements regarding the correct positioning of the eyeball.

Three studies reported post-operative bone contact between the bony segments and the native bone. An average of 88.1% of cases had bone to bone contact [11,26,27].

## 4. Discussion

This systematic review on the accuracy of CAS in maxillary reconstruction demonstrated a lack of uniformity in image acquisition, maxillary defect classification, and postoperative evaluation methodologies, which limits legitimate comparisons of postoperative accuracy results between studies. No meta-analyses were feasible because of differences in the postoperative measurement methods, despite the fact that all of the included studies focused on the positioning of bony segments. Only approximate comparisons of postoperative accuracy between studies were feasible; the accuracy of the fibular segments in the included studies ranged between 0.44 mm and 7.8 mm and between 2.90° and 6.96° (Table 3). Eight studies reported deviation of the bony segments in mm, of which seven studies described deviations < 5 mm and one study described deviations < 10 mm. Two studies reported that 100% of bony segments were placed in the planned anatomical position. This largely confirms our hypothesis that postoperative results remain within 5 mm compared to the preoperative virtual plan when CAS is used, but in one case a bigger deviation of 7.8 mm was noticed. Therefore, we cannot take it for granted that CAS is better than traditional free hand surgery. Thus, further research is needed to (1) determine if CAS is truly more accurate than traditional free hand surgery and (2) to determine the range of deviation without consequences for the postoperative function of primary placed dental implants and the eyeball.

Eight of the included studies were case series, which represent low-quality evidence because of the lack of randomization, blinding, and a control group. Four of the included studies were case-control studies, also lacking randomization and blinding. Thus, the risk of bias in the included studies is indicated as being high. However, no postoperative accuracy measurements were compared between studies and no meta-analysis was feasible, which limits the influence of this high risk of bias on the interpretation of the results of this systematic review. Because of the limited amount of included studies and the lack of randomized controlled trials, the current evidence level on the accuracy of CAS maxillary reconstruction still needs to be considered as being low.

The accuracy of the postoperative result in maxillary reconstruction using CAS is interesting to investigate for several reasons. CAS seems to be superior to conventional free-hand surgery because of the more accurate postoperative results, but true evidence is lacking, even though three case control studies in our review confirmed superior outcomes of CAS compared with traditional free hand surgery [24,26,27]. Hypothetically, a high accuracy level of the reconstructed maxillary bony tissue has advantages in the functional outcome and quality of life after surgery: (1) CAS software helps to harvest only the necessary bone by sawing the precise sizes of the bony segments, which lowers donor site comorbidity when a DCIA is harvested [28,29]; (2) higher accuracy due to the preoperative virtual planning and the 3D printed guides used during surgery influences the tumor margin control during ablative surgery [30,31]; (3) an accurate postoperative inter-maxillary relationship compared with the virtual plan increases the likelihood of a functional prosthetic fit [32,33]; (4) insertion of dental implants can be performed during the reconstruction, potentially saving the patient from at least one extra surgical procedure [33,34]; and (5) correct position of the eyeball after reconstruction of the orbital floor is needed for both functional and aesthetic reasons. These suggestions are mostly based on research in mandibular reconstruction using CAS [13]. However, in maxillary reconstruction using CAS, these hypotheses and questions still need to be answered.

It is plausible that CAS will play an increasingly important role in reconstructive surgery in the future. CAS obviously offers new possibilities in the evaluation of postoperative results. All of the included studies compared the postoperative STL file with the preoperative STL (revised to the virtual plan). The goals of maxillary reconstruction (obliterate the defect, restore oral function, support the midfacial elements, and to restore the aesthetics of the face) all depend on the correct positioning of the bony segments during surgery. The comparison of the postoperative STL file with the preoperative STL file (revised to the virtual plan) seems to be the most relevant evaluation method to evaluate these bony segment positions. Based on the studies included in this systematic review, we can state that there is consensus about this matter, but care must be taken to the 3D volumes of both STL models, which need to be similar [12]. Table 3 shows that only 6/12 studies described the measurement software type. Reconstructive surgeons should be able to repeat the evaluation method on their own reconstructions so as to compare postoperative results. Therefore, the measurement software should always be mentioned in the materials and methods section of studies regarding postoperative accuracy in maxillary reconstruction using CAS.

As stated before by Brown and Shaw, the published accuracy results in this study are hard to interpret as level one or two evidence for the management of maxillary and midface defects does not exist [2]. Besides this, most of the included studies are descriptive, the total number of cases is low, the pathology is varied, the defect sizes differ, and different postoperative evaluations are performed; thus, the value of the included case-control studies is limited. Even with a standardized evaluation method, validated comparisons of postoperative accuracy results between studies are challenging. Notwithstanding these limitations, this study suggests that to improve the management of maxillectomy defects, the postoperative accuracy evaluation needs to be performed in an identical approach between studies as a first step. Based on the identical evaluation of results between studies, different approaches and techniques in maxillary reconstruction can be compared. In addition, the European Union medical device regulation (MDR) requires a standardization and Conformité Européenne (CE) certification for all CAS phases [35]. There is thus a definite need for a standardized postoperative evaluation guideline in maxillary reconstruction using CAS.

To compare reconstructions with a similar size and complexity, a classification of the maxillary and midface defect is indispensable. Of the studies included in this systematic review, 7/12 used the Brown maxillary and midface defect classification (Table 1), which describes a vertical and horizontal component of the defect [2]. The included studies make no attempt to differentiate accuracy results between different types of Brown classes and often average the accuracy results of their cohort (including different Brown classes; Table 3). It is recommended for authors to use and mention the Brown maxillary and midface defect class in future studies in order to facilitate legitimate comparisons between reconstructions with the same complexity. Because the total number of cases in maxillary reconstruction using CAS worldwide is low, even in the biggest cohorts, it makes sense to separate accuracy results per Brown class and to report the results per case, otherwise no conclusions regarding the accuracy are meaningful.

In addition, the evaluation methods need to be clear and reproducible. Anatomical position, used as an accuracy parameter by Navarro Cuéllar et al. [27] and Swendseid et al. [26], is poorly defined. Deviations measured in millimeters and degrees are far more interpretable, making future meta-analyses more feasible.

A Brown class III or IV maxillary defect including the orbital floor, which needs to be reconstructed for the correct position of the eyeball, is reported in 5/12 of the included studies. However, in only 3/12 studies is the management of the orbital floor reconstruction mentioned in the materials and methods section (reconstructed with a pre-bent titanium mesh or a 3D printed patient specific titanium mesh). No postoperative quantitative measurements regarding the orbital floor or the position of the eyeball are reported.

In the footsteps of our evaluation guideline for mandibular reconstruction using CAS [12,14], we will propose a practical, feasible, and reproducible evaluation guideline for maxillary and midface reconstructions using CAS in the future. This offers possibilities to study the influence of the parameters mentioned in Table 1, Table 2 and Table 3 on the postoperative accuracy results, which could provide more definitive evidence regarding the management of maxillary and midface defects. In addition, with the help of an evaluation guideline, further research can be set up to investigate the longitudinal follow-up of functional outcomes, aesthetics, quality of life, and costs, separated for all different Brown classes in maxillary reconstruction using CAS. This will also prove if CAS really contributes a significant difference over the conventional methods.

## 5. Conclusions

This systematic review on the accuracy of CAS in maxillary reconstruction showed heterogeneity in the image acquisition, maxillary defect classification, and postoperative evaluation methods between the studies in the current literature, which limits legitimate comparisons of postoperative accuracy results between studies. As a first step, a postoperative evaluation guideline to create uniformity in evaluation methods needs to be considered in order to allow for valid comparisons of postoperative results and to facilitate meta-analyses in the future. With this proper validation of postoperative results, future research might explore more definitive evidence regarding the management and superiority of CAS in maxillary and midface reconstruction.

## Figures and Tables

**Figure 1 jcm-10-01226-f001:**
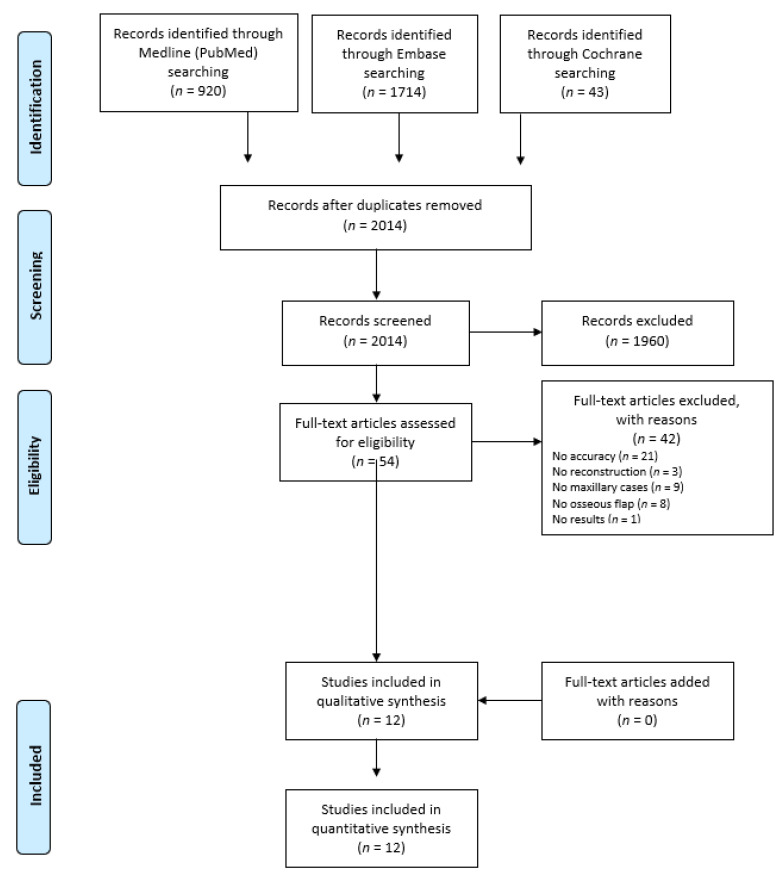
Flowchart methodology for the study selection process.

**Table 1 jcm-10-01226-t001:** Characteristics of the included studies.

Author	No. CAS Cases (*n* = 67)	Reconstruction	Defect Size	Flap Type	No. Segments (Class)	Plate Type	Orbital Floor Reconstruction
Liu et al. (2009) [18]	2	Primary (1) and secondary (1)	Lower maxilla (2)	FFF (2)	N/A	N/A	N/A
Melville et al. (2019) [19]	1	Primary	Brown class IId	FFF	2	PSRP	No
Morita et al. (2017) [20]	1	Primary	N/A	FFF	3	Pre-bent mini-plates	No
Navarro Cuéllar et al. (2021) [27]	6 (vs. 6 control)	Primary	Brown class IIc	FFF (6)	3	Pre-bent mini-plates	No
Numajiri et al. (2018) [21]	4	Primary (2) and secondary (2)	N/A	FFF (4)	N/A	Pre-bent mini-plates	Yes (1), pre-bent TM
Schepers et al. (2016) [8]	5	Secondary (5)	Partial maxillectomy (1)	FFF (5)	3, 2, and 1 (partial maxillectomy)	Mini-plates	No
Swendseid et al. (2019) [26]	9 (vs. 14 control)	Primary	Brown classes II (2), III (2), IV (1), and V (4)/Cordeiro classes II (2), IIIa (2), IIIb (3), and IV (2)	SOFF (9)	1.8 (mean)	PSRP (3) Mini-plates	N/A
Tarsitano et al. (2016) [22]	4	Primary (4)	Brown classes II (2) and III (2)	FFF (4)	2 (Brown classes II and III)	PSTM	Yes (2), 3D printed PSTM
Wang et al. (2016) [11]	18	Primary (18)	Brown classes I (1), II (9), III (5), and IV (3)	FFF (18)	N/A (mean 2.8 ± 0.91)	Pre-bent mini-plates	N/A
Yang et al. (2018) [23]	3	Secondary (3)	Left maxilla (1), right maxilla (1), and anterior maxilla (1)	FFF (3)	3 (left maxilla), 1 (right maxilla), and 2 (anterior maxilla)	PSRP	N/A
Zhang et al. (2015) [24]	8 (vs. 19 control)	Primary (8)	Brown classes II (5) and III (3)	FFF (8)	N/A	Pre-bent mini-plates	Yes (3), pre-bent TM
Zheng et al. (2016) [25]	6	Primary (4) and secondary (2)	Brown classes Ib, IIc, IId, IIIb, IIId, and IVb	FFF (6)	1 (Brown class Ib), 3 (Brown classes IIc, IId, IIIb, IIId, and IVb)	N/A	N/A

CAS—computer-assisted surgery; FFF—fibula free flap; SF—subscapular system free flap; N/A—not available; PSRP—patient-specific reconstruction plate; TM—titanium mesh; PSTM—patient specific titanium mesh.

**Table 2 jcm-10-01226-t002:** CAS process of the included studies.

Author	No. CAS Cases (*n* = 67)	Pre-op Craniofacial Imaging	Pre-op Donor Site Imaging	CAS Software	Mirroring Tool	3D Printed Devices	CAS Dental Implants	Surgical Navigation	Post-op Imaging
Liu et al. (2009) [18]	2	CT (FOV 20cm, pitch 1.0, 0.75 mm ST, 120–280mA)	N/A	Surfacer ^1^	N/A	Model neomaxilla	No	No	CT
Melville et al. (2019) [19]	1	CT	CT	Proplan CMF ^2^	N/A	Cutting guides maxilla Cutting guide fibula	No	No	CBCT
Morita et al. (2017) [20]	1	N/A	N/A	Free software	N/A	Cutting guide fibula Model osteotomized fibula Model osteotomized maxilla	No	No	CT
Navarro Cuéllar et al. (2021) [27]	6 (vs. 6 control)	CT	CT	ProPlan CMF ^2^	No	Cutting guide fibula Cutting guide maxilla Model neomaxilla	No	No	CT
Numajiri et al. (2018) [21]	4	N/A	N/A	InVesalius ^3^	N/A	Cutting guide fibula Cutting guides maxilla Model native maxilla Model native fibula	No	No	CT
Schepers et al. (2016) [8]	5	CBCT (120 kV, 5 mA, 0.4 voxel, FOV 23 × 16 cm)	CTA (0.6 mm collimation, 30 f kernel)	Proplan CMF 1.3 ^2^, Simplant Pro 2011 ^2^, 3-matic 7.0 ^2^	N/A	Cutting guide fibula Cutting guides maxilla Drill guide dental implants Model neomaxilla Titanium bridge abutment	Yes (5)	No	CBCT
Swendseid et al. (2019) [26]	9 (vs. 14 control)	N/A	N/A	ProPlan CMF ^2^, Stryker CMF ^5^, IPS CaseDesigner ^7^	Yes (1)	Cutting guide scapula Cutting guide maxilla Model neomaxilla	Yes (1)	No	N/A
Tarsitano et al. (2016) [22]	4	CT (0.6 mm ST, 120 kV, 225 mA, 0.5 pitch, 1 s rotation time, 0.6 mm collimation, 512 × 512 matrix size)	CTA (soft tissue kernel 2.5 mm ST)	Rhino 4.0 ^4^	Yes	Cutting guide fibula Cutting guides maxilla	No	Yes, Nlite Stryker ^5^	CT
Wang et al. (2016) [11]	18	CT	CT	Proplan CMF ^2^	N/A	Cutting guide fibula Cutting guides maxilla Model neomaxilla Shape template neomaxilla	No	No	CT
Yang et al. (2018) [23]	3	CT	CT	Mimics ^2^, Proplan CMF ^2^	N/A	Cutting guide fibula Cutting guides maxilla	No	No	CT
Zhang et al. (2015) [24]	8 (vs. 19 control)	CT (FOV 20 cm, pitch 1.0, 0.7 5 mm ST, 120Y280mA)	CT (FOV 20cm, pitch 1.0, 0.75 mm ST, 120Y280mA)	iPlan CMF ^2^, Proplan CMF ^2^	Yes	Model mirrored maxilla Shape template neomaxilla	No	Yes, Brainlab ^6^	CT
Zheng et al. (2016) [25]	6	CT	CT	Mimics ^2^	N/A	Cutting guide fibula Cutting guides maxilla Shape template neomaxilla	No	No	N/A

Abbreviations: CAS, computer-assisted surgery; Pre-op, preoperative; Post-op, postoperative; CT, computed tomography; FOV, field of view; ST, slice thickness; N/A, not available; CBCT, cone beam computed tomography; CTA, computed tomography angiography; kV, kilovolt; mA, milliampere; ^1^ EDS Company, Plano, Texas, USA; ^2^ Materialise NV, Leuven, Belgium; ^3^ Center for Information Technology Renato Archer—CTI, Campinas, Brazil; ^4^ Robert McNeel & Associates, Seattle, WA, USA; ^5^ Stryker, Kalamzoo, MI, USA; ^6^ Brainlab, Feldkirchen, Germany; ^7^ KLS Martin group, Jacksonville, FL, USA.

**Table 3 jcm-10-01226-t003:** Postoperative accuracy measurements.

Author	No. CAS Cases (*n* = 67)	Measurement Software	Comparison	Methodology	Results
Liu et al. (2009) [18]	2	N/A	STL post-op vs. STL pre-op revised	Deviation fibula	Mandibular and maxillary results merged
Melville et al. (2019) [19]	1	N/A	STL post-op vs. STL pre-op revised	Anterior and posterior width (neo)maxilla difference	2.2 mm/7.8 mm
				Fibular segment dimension differences:	
				Height posterior section	0.9 mm
				Height anterior section	1.6 mm
				Height medial section	2.0 mm
				Greatest discrepancy fibula	2.2 mm
Morita et al. (2017) [20]	1	N/A	STL post-op vs. STL pre-op revised	Deviation fibula	2–4 mm
Navarro Cuéllar et al. (2021) [27]	6 (vs. 6 control)	N/A	STL post-op vs. STL pre-op revised	CAS vs. control	
				Anatomical position of bone (%)	100% vs. 66.6% (*p* = 0.028)
				Bone contact (%)	100% vs. 83.3% (*p* = 0.041)
				Change of vertical distance (mm)	3.28 ± 1.43 vs. 6.73 ± 2.14 (*p* = 0.019)
				Horizontal shift > 5 mm (%)	16.6% vs. 83.3% (*p* = 0.016)
Numajiri et al. (2018) [21]	4	CloudCompare ^1^	STL post-op vs. STL pre-op revised	Average deviation fibula	0.44 mm
Schepers et al., (2016) [8]	5	Geomagic Studio ^2^	STL post-op vs. STL pre-op revised	Fibula segments as reference:	
				Mean center point deviation fibula segments	0.93 mm
				Mean angulation deviation fibula segments	2.90°
				Mean center point deviation implants	1.93 mm
				Mean angulation deviation implants	3.67°
				Occlusion as reference:	
				Mean center point deviation fibula segments	5.41 mm
				Mean angulation deviation fibula segments	6.96°
				Mean center point deviation implants	4.95 mm
				Mean angulation deviation implants	6.26°
Swendseid et al. (2019) [26]	9 (vs. 15 control)	ProPlan CMF ^4^	STL post-op vs. STL pre-op revised	CAS vs. control	
				Anatomical position of bone (%)	100% vs. 71% (*p* = 0.035)
				Bone contact (%)	70% vs. 60% (*p* = 0.49)
				CAS	
				Mean position deviation (mm)	7.2 mm
				Position deviation < 10 mm (%)	82%
Tarsitano et al. (2016) [22]	4	GOM ^3^ SimPlant ^2^	STL post-op vs. STL pre-op revised	Average deviation fibula + titanium mesh	1.1 mm
Wang et al. (2016) [11]	18	N/A	STL post-op vs. STL pre-op revised	CAS vs. control	
				Overextension of horizontal ends of fibular segments (n)	1 (5.6%)
				Overextension of vertical ends of fibular segments (n)	1 (5.6%)
				Precise bone-to-bone contact (n)	17 (94.4%)
Yang et al. (2018) [23]	3	Mimics ^4^	STL post-op vs. STL pre-op revised	Angulation deviation bone grafts Distance deviation bone grafts Mean absolute distance deviation	Mandibular and maxillary results merged
Zhang et al. (2015) [24]	8 (vs. 19 control)	Geomagic Studio ^2^	STL post-op vs. STL pre-op revised	CAS vs. control	
				Change of vertical distance	2.82 mm vs. 6.13 mm
				Horizontal shift (>5 mm) (n)	2 (25%) vs. 14 (73.6%)
				Overextension of the posterior end of the fibula (n)	1 (12.5%) vs. 10 (52.6%)
Zheng et al. (2016) [25]	6	N/A	STL post-op vs. STL pre-op revised	Average central point deviation	0.58 mm
				Maximum deviation	1.53 mm
				Rotation	6.0°

Abbreviations: CAS, computer-assisted surgery; N/A, not available; STL, standard tessellation language; Post-op, postoperative; Pre-op, preoperative; DICOM, Digital Imaging and Communications in Medicine; ^1^ Free open-source; ^2^ Geomagic, Morrisville, NC, USA; ^3^ GOM mbH, Braunschweig, Germany; ^4^ Materialise NV, Leuven, Belgium.

## Data Availability

Not applicable.

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
