# Peer review of "Accuracy of Computer-Assisted Surgery in Maxillary Reconstruction: A Systematic Review"

_jcm, 2021, doi:10.3390/jcm10061226_

Round 1

Reviewer 1 Report

Introduction

  1. The accuracy of CAS maxillary reconstruction is a very general description. It's the key of this study however the authors did not provide enough information in the introduction section. What is the accuracy of CAS maxillary reconstruction? The accuracy of graft harvesting? The Accuracy of recipient site preparation? The accuracy of placed graft vs digital plan? The accuracy of placed implant position vs digital plant? Please explain more about what accuracy you are going to study in the introduction section.

Material and methods

  1. Please indicate the types of included research you were aiming to include in the Material and Method section. Prospective? retrospective? case series? cohort or randomized controlled clinical trials?

  1. The aim of this study is to assess the accuracy of CAS maxillary reconstruction. Please indicate what measures (primary outcome) are used to assess accuracy.

Discussion

  1. Please discuss the current evidence level on the accuracy of CAS maxillary reconstruction.

  1. Please discuss what kind of research is needed in the field in the future.

Author Response

Responses to comments of reviewer 1

Accuracy of Computer-Assisted Surgery in Maxillary Reconstruction:
A Systematic Review

Manuscript ID: jcm-1149616

We wish to thank the reviewer for the kind comments that enabled us to improve our systematic review. We have earnestly tried to clarify the unclear points that were identified.

Introduction

The accuracy of CAS maxillary reconstruction is a very general description. It's the key of this study however the authors did not provide enough information in the introduction section. What is the accuracy of CAS maxillary reconstruction? The accuracy of graft harvesting? The Accuracy of recipient site preparation? The accuracy of placed graft vs digital plan? The accuracy of placed implant position vs digital plant? Please explain more about what accuracy you are going to study in the introduction section.

We thank the reviewer for mentioning this gap in the introduction.

We added the following sentences to the last section of the introduction (Page 2):

The accuracy evaluation of the postoperative bony maxillary reconstruction (including potential CAS guided dental implants) versus the preoperative virtual plan provides the operator with information about the execution of the operation and gives an opportunity to compare results with performed (CAS) techniques from other studies.

We added some words to the following sentences for a better reading flow:

However, uniformity in accuracy evaluation methods is lacking and no evaluation guide-line like the one previously published for mandibular reconstruction using CAS exists for maxillary reconstructions yet [12].

Similar to our previously published systematic review on accuracy in mandibular recon-struction [13,14], we reviewed all studies that quantitatively assessed accuracy of maxillary reconstruction performed with CAS as mentioned above.

Material and methods

Please indicate the types of included research you were aiming to include in the Material and Method section. Prospective? retrospective? case series? cohort or randomized controlled clinical trials?

We thank the reviewer for noting this.

We added the following sentences to the materials and methods section (Page 3):

Randomized controlled trials, cohort studies (prospective observational), case-control studies and case reports and series (either prospective or retrospective) were all considered applicable to this systematic review.

The aim of this study is to assess the accuracy of CAS maxillary reconstruction. Please indicate what measures (primary outcome) are used to assess accuracy.

We thank the reviewer for mentioning this gap in the results section.

We added some words to the following sentence in the results section (Page 7):

In all cases a postoperative STL model was compared to a preoperative STL model (revised to the virtual plan) and the postoperative accuracy primary outcome measurements mainly focused on the position of the bony segments (deviation (mm), shift (mm), dimension differences (mm), center point deviation (mm), or rotation (degrees)). The primary outcome measurements of the bony segments in the included studies ranged between 0.44mm and 7.8mm and between 2.90° and 6.96°.

Discussion

Please discuss the current evidence level on the accuracy of CAS maxillary reconstruction.

We thank the reviewer for this comment.

We added the following sentence to the second paragraph of the discussion section (Page 8):

Due to the limited amount of included studies, and the lack of randomized controlled trials, the current evidence level on the accuracy of CAS maxillary reconstruction still needs to be considered as low.

Please discuss what kind of research is needed in the field in the future.

We thank the reviewer for this comment.

We added the following sentence to the last paragraph of the discussion section (Page 10):

In addition, with the help of an evaluation guideline further research can be set up to investigate the longitudinal follow-up of functional outcomes, aesthetics, quality of life and costs, separated for all different Brown classes in maxillary reconstruction using CAS. This also will proof if CAS really contributes to a significant difference over the conventional methods.

Reviewer 2 Report

Dear Authors,

I have read carefully your systematic review on Accuracy of computer-assisted surgery in maxillary reconstruction.

The analyzed issue is a very stimulating topic for oral and maxillo facial surgeons. The methodology is sound and the presented results, within the limitations of the study, contribute to fulfill the knowledge gap on Computer Assisted Surgery (CAS). Discussion and Conclusion sections are appropriate and highlight the limitations of the study. For all these reasons I have really appreciated your efforts and consequently, in my opinion, the submitted paper fully meets the quality standard required by Journal of Clinical Medicine in the present form.

Author Response

We thank the reviewer for the kind comments.

Reviewer 3 Report

This Systematic Review deals with the accuracy of CAS surgery in the reconstruction of maxillary defects. The study is properly designed and well structured. As described by the authors, the problem is the difficult comparability of the imaging techniques used as well as the software programs and their segmentation methods, and the measurement methods ultimately used. Standardization of the CAS process as well as comparison of the software / imaging techniques as well as the measurement methods should improve the results in the future. The work is to be published as it is.

Author Response

(The authors gave the same response as above.)
